**Data Availability Statement:** We understand the importance of openness of data. However, unfortunately, the data for this manuscript cannot

# Real-world safety and effectiveness of rivaroxaban using Japan-specific dosage during long-term follow-up in patients with atrial fibrillation: XAPASS

**Takanori Ikeda[1]***, **Satoshi Ogawa[2], Takanari Kitazono[3], Jyoji Nakagawara[4,5], Kazuo Minematsu[5,6], Susumu Miyamoto[7], Yuji Murakawa[8], Sanghun Iwashiro[9], Yutaka Okayama[9], Toshiyuki Sunaya[10], Kazufumi Hirano[9], Takanori Hayasaki[9]**

1 Department of Cardiovascular Medicine, Toho University Graduate School of Medicine, Tokyo, Japan, 2 International University of Health and Welfare, Mita Hospital, Tokyo, Japan, 3 Department of Medicine and Clinical Science, Graduate School of Medical Sciences, Kyushu University, Fukuoka, Japan, 4 Osaka Namba Clinic, Osaka, Japan, 5 National Cerebral and Cardiovascular Center, Suita, Japan, 6 Iseikai Medical Corporation, Osaka, Japan, 7 Department of Neurosurgery, Kyoto University Graduate School of Medicine, Kyoto, Japan, 8 The 4th Department of Internal Medicine, Teikyo University School of Medicine, Mizonokuchi Hospital, Kawasaki, Japan, 9 Medical Affairs and Pharmacovigilance, Bayer Yakuhin, Ltd., Osaka, Japan, 10 Research & Development Japan, Bayer Yakuhin, Ltd., Osaka, Japan

* takanori.ikeda@med.toho-u.ac.jp

## Abstract

### Background

The Xarelto Post-Authorization Safety and Effectiveness Study in Japanese Patients with Atrial Fibrillation (XAPASS) was designed to investigate safety and effectiveness during long-term follow-up of rivaroxaban treatment, using reduced doses compared with other global regions, in Japanese patients with non-valvular atrial fibrillation in real-world clinical practice.

### Methods

In this prospective, open-label, single-arm, observational study, 11,308 patients with non-valvular atrial fibrillation newly prescribed rivaroxaban (15/10 mg once daily) at 1416 sites across Japan were enrolled and followed for a mean of 2.5 years.

### Results

In total, 10,664 and 10,628 patients were included in the safety and effectiveness analyses, respectively. In the safety population, mean (standard deviation) age was 73.1 (9.8) years and Congestive heart failure, Hypertension, Age ≥75 years, Diabetes mellitus, previous Stroke/TIA (2 points) ($CHADS_2$) score was 2.2 (1.3). Incidences (95% confidence intervals) of any and major bleeding were 3.77 (3.53–4.01) and 1.16 (1.03–1.29) events per 100 patient-years, respectively. Age ≥75 years, creatinine clearance <50 mL/min, diabetes mellitus, and vascular disease were independently associated with incidence of major bleeding. The primary composite effectiveness outcome of stroke, non-central nervous system systemic embolism, and myocardial infarction occurred at an incidence (95% confidence interval) of 1.32 (1.18–1.46) events per 100 patient-years. Age ≥75 years, hypertension, prior

be made publicly available, because it was collected from the participating medical facilities and public availability would compromise patient confidentiality or participant facility privacy. These restrictions are compliant with the Ethical Guidelines for Medical and Health Research Involving Human Subjects that was issued by Ministry of Health, Labour and Welfare (MHLW) of Japan and with the Good Post-marketing Surveillance Practice, which was also issued by MHLW of Japan. The persons who are responsible for the data request are Sanghun Iwashiro (sanghun iwashiro@bayer.com), Yutaka Okayama (yutaka.okayama@bayer.com) or Kazufumi Hirano (kazufumi.hirano@bayer.com).

**Funding:** XAPASS is a post-marketing surveillance study funded by Bayer Yakuhin, Ltd. (Osaka, Japan). A steering committee (S1 Appendix) was responsible for developing the protocol and the case report form and for oversight of both the conduct of the study and the database, and is accountable for analysis and publication of the results. Bayer Yakuhin, Ltd provided operational oversight of the study, prepared data and the manuscript, and paid the salaries of its employees (SI, YO, TS, KH, and TH).

**Competing interests:** TI, SO, TK, JN, KM, SM, and YM have been advisory board members for Bayer Yakuhin, Ltd. TI has received research grants from Bay er Yakuhin, Ltd., Bristol-Myers Squibb, Daiichi Sankyo, Medtronic Japan, and St. Jude Medical, and honoraria from Bayer Yakuhin, Ltd., Bristol-Myers Squibb, Daiichi Sankyo, Ono, and Pfizer, and was an advisory board member for Bristol-Myers Squibb. TK has received a research grant from Bayer Yakuhin, Ltd. JN has received a research grant from Nihon Medi-Physics. KM has received honoraria from Astellas, AstraZeneca, Bayer Yakuhin, Ltd., Boehringer Ingelheim, Bristol-Myers Squibb, Daiichi Sankyo, Japan Stryker, Kowa, Mitsubishi- Tanabe, Nihon Medi-Physics, Nippon Chemiphar, Otsuka, Pfizer, Sawai, and Sumitomo Dainippon, and was an advisory board member for CSL Behring and Medico's Hirata. SM has received research grants from Astellas, Brainlab, Bristol-Myers Squibb, Carl Zeiss Meditec, Chugai, CSL Behring, Daiichi Sankyo, Eisai, Medtronic, Meiji, Mitsubishi-Tanabe, Mizuho, MSD, Nihon Medi-Physics, Otsuka, Pfizer, Philips Electronics Japan, Sanofi, Siemens Healthcare, and Takeda. YM has received research grants from Bayer Yakuhin, Ltd., Boehringer Ingelheim, and Daiichi Sankyo, and honoraria from Bayer Yakuhin, Ltd., Boehringer Ingelheim, Bristol-Myers Squibb, and Daiichi Sankyo. SI, YO, TS, KH, and TH are employees of Bayer Yakuhin, Ltd. This does not alter our

ischemic stroke/transient ischemic attack, and concomitant use of antiplatelets were independently associated with incidence of the composite outcome of stroke, non-central nervous system systemic embolism, and myocardial infarction.

## Conclusion

In the XAPASS, a large-scale study involving a broad range of patients with non-valvular atrial fibrillation newly prescribed rivaroxaban using Japan-specific dosage in real-world clinical practice, no unexpected safety or effectiveness concerns were detected during up to 5 years of follow-up.

## Introduction

Atrial fibrillation (AF) is the most common arrhythmia in clinical practice and the most common cause of ischemic stroke, the risk of which in patients with AF is highly variable depending on patients' characteristics and comorbidities [1–3]. Clinical guidelines therefore recommend anticoagulant therapy in patients with AF considered to be at risk of stroke [4–6]. Vitamin K antagonists have well-established efficacy in reducing the risk of stroke in these patients, but their use in real-world clinical practice is limited by common food and drug interactions, and their narrow therapeutic index, which necessitates regular coagulation monitoring and dose adjustments [5]. Four direct oral anticoagulants (DOACs)—dabigatran, rivaroxaban, apixaban, and edoxaban—have now been approved worldwide and are recommended in clinical guidelines for the prevention of stroke and systemic embolism (SE) in patients with non-valvular AF (NVAF) [4–6]. With a rapid onset of action, fewer food and drug interactions, and predictable effects with no requirement for regular monitoring and dose adjustment, these DOACs are now widely used in clinical practice [5, 6].

The DOAC rivaroxaban is a direct factor Xa inhibitor and was shown to be non-inferior to warfarin in the prevention of stroke/SE in patients with NVAF and to be associated with a similar risk of major bleeding and a lower risk of intracranial and fatal bleeding, in a global phase 3 clinical trial [7]. In a phase 3 Japanese study, rivaroxaban dosages were reduced compared with the dosage used in other global regions, based on the different pharmacokinetic and pharmacodynamic characteristics of rivaroxaban between Japanese patients and non-Japanese patients [8]. The Japan-specific dosage of rivaroxaban was found to be non-inferior to warfarin with respect to the incidence of major and non-major clinically relevant bleeding, and was associated with a lower incidence of intracranial bleeding and a strong trend for a lower incidence of stroke/SE compared with warfarin [9].

Phase 3 trials provide robust evidence of a drug's safety and efficacy in the selected patient population under controlled conditions and form the basis of regulatory approvals. However, post-authorization studies are required to evaluate the safety and effectiveness of treatments in a broader range of patients treated in routine clinical practice. The Xarelto Post-Authorization Safety and Effectiveness Study in Japanese Patients with Atrial Fibrillation (XAPASS; NCT01582737) was a prospective, real-world observational study designed to establish the safety and effectiveness of rivaroxaban in clinical practice in Japan and was mandated by the Japanese regulatory authority as post-marketing surveillance [10]. An analysis of 1-year outcomes in the XAPASS, including major bleeding and thromboembolic events, indicated that rivaroxaban has a favorable safety profile and is effective in preventing stroke in Japanese patients in clinical practice [11]. The safety and effectiveness of rivaroxaban in these Japanese patients in the XAPASS are now reported for the entire follow-up period of up to 5 years.

adherence to PLOS ONE policies on sharing data and materials.

## Methods

### Design

The design of the XAPASS has been described previously [10, 11]. In brief, the XAPASS (NCT01582737) was a prospective, open-label, single-arm, observational, non-interventional, cohort study conducted to confirm the safety and effectiveness profiles of rivaroxaban in real-world use in Japan across a wide range of patients with NVAF. Patients were enrolled at 1416 sites across Japan from April 2012 to June 2014. The standard observation period was 2 years for each patient, with data collection at 6 months, 1 year, and 2 years after rivaroxaban initiation; annual follow-up assessments were then continued for a maximum of 5 years. Information on all observed events was recorded in case report forms. For patients in whom rivaroxaban therapy was discontinued, observation was continued for a further 30 days.

The design of this study was reviewed and approved by the Pharmaceuticals and Medical Devices Agency; it was conducted in accordance with the standards for Good Post-Marketing Study Practice provided by the Ministry of Health, Labour and Welfare in Japan and conforms to the ethical guidelines of the 1975 Declaration of Helsinki. Separate ethics approval for this post-marketing surveillance study and written informed consent to participate in the surveillance were not required under standards for Good Post-Marketing Study Practice, but ethical approvals from institutional review board in participating center were obtained if required by each center. Any author could not access to patients' medical records in this study.

### Patients

Men and women with NVAF who were starting rivaroxaban therapy to reduce the risk of stroke/SE were included in the study. Contraindications to rivaroxaban were considered according to the Japanese product label.

### Treatment

In Japan, rivaroxaban is approved at doses of 15 mg and 10 mg once daily, which are lower than the dosage used in other countries for patients with a creatinine clearance (CrCl) of $\geq$50 mL/min and <50 mL/min, respectively. In the XAPASS, physicians were recommended to prescribe appropriate dosage of rivaroxaban in accordance with Japanese package insert. The duration of rivaroxaban treatment was determined at the physicians' discretion.

### Outcomes

All outcomes were recorded as adverse events. The primary safety outcome was rates of any bleeding, including major bleeding, as defined using International Society on Thrombosis and Haemostasis criteria [12], and non-major bleeding. The primary effectiveness outcome was a rate of composite of stroke, non-central nervous system SE, and myocardial infarction (stroke/non-CNS SE/MI). Definitions of the outcomes have been described previously [10]. Intracerebral bleeding was reported as both a stroke and a bleeding event. Transient ischemic attacks (TIAs) were not included in the stroke outcome.

Secondary outcomes included all-cause mortality and the rates of adverse events (bleeding and stroke/non-CNS SE/MI) in patients with different baseline scores for $CHADS_2$ (Congestive heart failure, Hypertension, Age $\geq$75 years, Diabetes mellitus, previous Stroke/TIA [2 points]), $CHA_2DS_2$-VASc (Congestive heart failure, Hypertension, Age [65–74 years, 1 point; $\geq$75 years, 2 points], Diabetes mellitus, previous Stroke/TIA [2 points], Vascular disease, and female sex) and modified HAS-BLED (Hypertension, Abnormal renal or liver function, previous Stroke, previous major or predisposition to Bleeding, Labile international normalized

ratio [excluded from this analysis], Elderly [>65 years], medication use predisposing to bleeding, and previous Drug or alcohol use).

All outcomes reported over the 5-year follow-up period were included in this analysis.

## Statistical analysis

The target number of patients for enrollment into this study was 10,000, based on the feasibility of recruitment.

Statistical analysis was performed using SAS version 9.2 or higher (SAS Institute Inc., Cary, NC). Data are presented as mean (standard deviation; SD) for continuous variables and as frequency (number and proportion of patients) for categorical variables. For the key safety and effectiveness outcomes, crude incidence (number and proportion of patients with the event) and incidence rate of patients with events per 100 patient-years, together with the corresponding 95% confidence intervals (CI), are presented. Kaplan–Meier plots were constructed to assess the time course and cumulative incidence of the safety and effectiveness outcomes. Hazard ratios and corresponding 95% CI were estimated using univariate and multivariate Cox regression analyses to assess the impact of patient characteristics on the outcomes; these characteristics included age ($\geq$75 years), sex (female), weight ($\leq$50 kg), renal function (CrCl <50 mL/min), prior ischemic stroke/TIA, concomitant use of antiplatelets, and comorbidities such as congestive heart failure, hypertension, diabetes mellitus, vascular disease (defined as MI and/or peripheral artery disease and/or aortic plaque), and liver dysfunction.

## Results

### Patients

In total, 11,308 patients were enrolled in the study, and 10,664 were included in the final safety analysis (Fig 1). Of the 644 patients omitted from the safety analysis, 531 were excluded owing to data not being collected. For the final analysis of the effectiveness of rivaroxaban, 10,628 patients were included; 34 patients were excluded owing to off-label use of rivaroxaban for

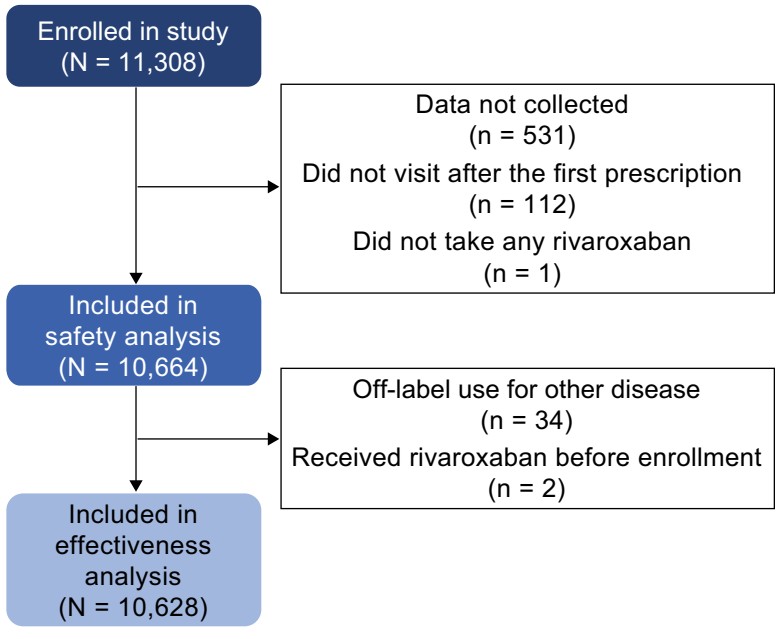

**Fig 1. Study flow chart.**

other diseases and two patients were excluded for receiving rivaroxaban prior to enrollment. Three patients with CrCl <15 mL/min, which is a contraindication to rivaroxaban treatment in Japan, were included.

In the safety analysis population, the mean (SD) age was 73.1 (9.8) years, with 48.7% (5198/ 10,664) of patients aged ≥75 years; 61.9% (6600/10,664) of patients were male (Table 1). The mean (SD) weight was 61.4 (13.1) kg, and 19.3% (2057/10,664) of patients were ≤50 kg. Creatinine clearance <50 mL/min was reported in 23.7% of patients. The mean (SD) CHADS$_2$ and CHA$_2$DS$_2$-VASc score was 2.2 (1.3) and 3.4 (1.6), respectively, and the mean (SD) modified HAS-BLED score was 1.5 (1.0). The most common comorbidity was hypertension (75.1% [8007/10,664]) followed by congestive heart failure (25.2% [2685/10,664]), prior ischemic stroke/TIA (23.0% [2450/10,664]), and diabetes mellitus (22.8% [2431/10,664]).

Patients received rivaroxaban for a mean (SD) of 891 (700) days (median, 739 days; range, 1–2458 days; interquartile range, 238–1483 days). A total of 5628/10,664 patients (52.8%) continued receiving rivaroxaban over the course of the study, and 2611/10,664 patients (24.5%) discontinued rivaroxaban treatment. The most common reasons for discontinuation were adverse events (44.0% [1,148/2,611]), patient's intention (13.4% [351/2,611]), death (12.3% [322/2,611]), and poor compliance (3.4% [89/2,611]). In total, 2425/10,664 patients (22.7%) were lost to follow-up, including patient transfers.

## Safety outcomes

Any bleeding (major and non-major bleeding) was reported in 963/10,664 patients (9.0%) and major bleeding in 307/10,664 patients (2.9%), corresponding to incidences (95% CI) of 3.77 (3.53–4.01) and 1.16 (1.03–1.29) events per 100 patient-years, respectively (Table 2). Intracranial hemorrhage occurred in 1.2% (123/10,664) of patients, with an incidence (95% CI) of 0.46 (0.38–0.54) events per 100 patient-years. The cumulative incidences of any bleeding and major bleeding increased over time (Fig 2).

The incidences of any bleeding, major bleeding, and intracranial hemorrhage were higher in patients with higher baseline CHADS$_2$, CHA$_2$DS$_2$-VASc, or modified HAS-BLED scores (S1 Table). The incidences of major bleeding according to baseline CHADS$_2$, CHA$_2$DS$_2$-VASc, and modified HAS-BLED scores are shown in S1 Fig.

In subgroup analyses, the incidence of major bleeding was greater with increasing age, weight ≤50 kg, renal impairment (CrCl <50 mL/min), congestive heart failure, diabetes mellitus, prior ischemic stroke/TIA, vascular disease, liver dysfunction, and concomitant antiplatelet therapy compared with the overall study population (S1 Table). In multivariate Cox regression analysis, age ≥75 years, CrCl <50 mL/min, diabetes mellitus, and vascular disease were independently associated with an increased incidence of major bleeding (Fig 3).

The crude incidence of all-cause mortality was 4.5% (475/10,664) and the incidence rate (95% CI) was 1.77 (1.61–1.93) events per 100 patient-years (Table 2).

Adverse events leading to death and any adverse events during the standard observation period (2 years) are detailed in S2 and S3 Tables, respectively.

## Effectiveness outcomes

The primary composite effectiveness outcome of stroke/non-CNS SE/MI occurred in 350/ 10,628 patients (3.3%), at an incidence (95% CI) of 1.32 (1.18–1.46) events per 100 patient-years. Stroke/non-CNS SE was reported in 316/10,628 patients (3.0%) (incidence [95% CI], 1.19 [1.06–1.32] events per 100 patient-years) and ischemic stroke in 227/10,628 patients (2.1%) (incidence [95% CI], 0.86 [0.74–0.97] events per 100 patient-years) (Table 2). The cumulative incidence of stroke/non-CNS SE/MI increased over time (Fig 2).

**Table 1. Characteristics of patients in the safety analysis population.**

| Characteristic | All patients (N = 10,664)[a] | |
|---|---|---|
| Age, years, mean (SD) | 73.1 (9.8) | |
| ≥65 | 8753 | 82.1 |
| ≥75 | 5198 | 48.7 |
| ≥85 | 1141 | 10.7 |
| Female sex | 4064 | 38.1 |
| Height, cm, mean (SD) | 160.1 (9.84) | |
| Body weight, kg, mean (SD) | 61.4 (13.05) | |
| ≤50 | 2057 | 19.3 |
| >50 | 7865 | 73.8 |
| Unknown | 742 | 6.96 |
| BMI, kg/m², mean (SD) | 23.9 (4.06) | |
| <18.5 | 589 | 5.5 |
| 18.5 to <25 | 4931 | 46.2 |
| 25 to <30 | 2403 | 22.5 |
| ≥30 | 551 | 5.2 |
| Unknown | 2190 | 20.5 |
| Serum creatinine, mg/dL, mean (SD) | 0.86 (0.70) | |
| Unknown | 140 | 1.3 |
| Creatinine clearance, mL/min, mean (SD) | 67.6 (26.3) | |
| <15 mL/min | 3 | 0.03 |
| 15 to <30 | 294 | 2.8 |
| 30 to <50 | 2233 | 20.9 |
| 50 to <80 | 4569 | 42.9 |
| ≥80 | 2727 | 25.6 |
| Unknown | 838 | 7.9 |
| $CHADS_2$ score, mean (SD) | 2.2 (1.3) | |
| 0 | 900 | 8.4 |
| 1 | 2601 | 24.4 |
| 2 | 3218 | 30.2 |
| 3 | 2093 | 19.6 |
| 4 | 1261 | 11.8 |
| 5 | 487 | 4.6 |
| 6 | 104 | 1.0 |
| $CHA_2DS_2$-VASc score, mean (SD) | 3.4 (1.6) | |
| 0 | 275 | 2.6 |
| 1 | 1002 | 9.4 |
| 2 | 1807 | 16.9 |
| 3 | 2498 | 23.4 |
| 4 | 2375 | 22.3 |
| 5 | 1566 | 14.7 |
| 6 | 786 | 7.4 |
| 7 | 298 | 2.8 |
| 8 | 54 | 0.5 |
| 9 | 3 | 0.03 |
| Modified HAS-BLED score, mean (SD)[b] | 1.5 (1.0) | |
| 0 | 1359 | 12.7 |
| 1 | 4523 | 42.4 |

(*Continued*)

**Table 1.** (Continued)

| Characteristic | All patients (N = 10,664)[a] | |
|---|---|---|
| 2 | 3270 | 30.7 |
| 3 | 1231 | 11.5 |
| 4 | 250 | 2.3 |
| 5 | 29 | 0.3 |
| 6 | 1 | 0.01 |
| 7 | 0 | – |
| 8 | 0 | – |
| Baseline comorbidities | | |
| Congestive heart failure | 2685 | 25.2 |
| Hypertension | 8007 | 75.1 |
| Diabetes mellitus | 2431 | 22.8 |
| Prior ischemic stroke/TIA | 2450 | 23.0 |
| Vascular disease[c] | 429 | 4.0 |
| Hepatic dysfunction | 722 | 6.8 |
| Type of AF | | |
| Paroxysmal | 3590 | 33.7 |
| Persistent | 3822 | 35.8 |
| Permanent | 2605 | 24.4 |
| Other | 27 | 0.3 |
| Unknown | 620 | 5.8 |
| Oral antiplatelet use | 474 | 4.4 |

[a] Data are presented as number and proportion of patients unless otherwise stated.

[b] Maximum score is 8 because the labile international normalized ratio was excluded.

[c] Vascular disease is defined as myocardial infarction and/or peripheral artery disease and/or aortic plaque.

Abbreviations: AF, atrial fibrillation; BMI, body mass index; CHADS$_2$, Congestive heart failure, Hypertension, Age ≥75 years, Diabetes mellitus, previous Stroke/TIA (2 points); CHA$_2$DS$_2$-VASc, Congestive heart failure, Hypertension, Age (65–74 years, 1 point; ≥75 years, 2 points), Diabetes mellitus, previous Stroke/TIA (2 points), Vascular disease, and female sex; Modified HAS-BLED, Hypertension, Abnormal renal or liver function, previous Stroke, previous major or predisposition to Bleeding, Labile international normalized ratio (excluded from this analysis), Elderly (>65 years), medication use predisposing to bleeding, and previous Drug or alcohol use; SD, standard deviation; TIA, transient ischemic attack.

The incidence of stroke/non-CNS SE/MI was higher in patients with higher CHADS$_2$, CHA$_2$DS$_2$-VASc, or modified HAS-BLED scores at baseline (S1 Fig; S4 Table).

Compared with the overall study population, the incidence of stroke/non-CNS SE/MI was greater with increasing age, weight ≤50 kg, CrCl <50 mL/min, congestive heart failure, prior ischemic stroke/TIA or vascular disease, and concomitant antiplatelet therapy (S4 Table). In multivariate Cox regression analysis, age ≥75 years, hypertension, prior ischemic stroke/TIA, and concomitant use of antiplatelets were independently associated with an increased incidence of stroke/non-CNS SE/MI (Fig 3).

## Outcomes according to dose

Of 10,664 patients included in the final safety analysis, 6953 (65.2%) started rivaroxaban at the recommended dose. The baseline demographic and disease characteristics of this subpopulation (S5 Table) were comparable with those of the overall study population (Table 1). Of 2872 patients who started rivaroxaban treatment at a non-recommended dose, 2594 patients

**Table 2. Safety and effectiveness outcomes.**

| | Crude incidence, n (%) | Incidence, events per 100 patient-years (95% CI) |
|---|---|---|
| **Safety outcomes (N = 10,664)** | | |
| Any bleeding | 963 (9.0) | 3.77 (3.53–4.01) |
| Major bleeding | 307 (2.9) | 1.16 (1.03–1.29) |
| Fatal bleeding | 35 (0.3) | 0.13 (0.09–0.17) |
| Critical organ bleeding | 136 (1.3) | 0.51 (0.42–0.59) |
| Intracranial hemorrhage | 123 (1.2) | 0.46 (0.38–0.54) |
| Hemoglobin decrease $\geq$2 g/dL | 118 (1.1) | 0.44 (0.36–0.52) |
| Transfusion of $\geq$2 units of packed red blood cells or whole blood | 46 (0.4) | 0.17 (0.12–0.22) |
| Non-major bleeding | 691 (6.5) | 2.68 (2.48–2.88) |
| All-cause mortality | 475 (4.5) | 1.77 (1.61–1.93) |
| **Effectiveness outcomes (N = 10,628)** | | |
| Stroke/non-CNS SE/MI | 350 (3.3) | 1.32 (1.18–1.46) |
| Stroke | 306 (2.9) | 1.15 (1.03–1.28) |
| Ischemic stroke | 227 (2.1) | 0.86 (0.74–0.97) |
| Hemorrhagic stroke | 88 (0.8) | 0.33 (0.26–0.40) |
| Non-CNS SE | 11 (0.1) | 0.04 (0.02–0.07) |
| MI | 35 (0.3) | 0.13 (0.09–0.17) |
| Stroke/non-CNS SE | 316 (3.0) | 1.19 (1.06–1.32) |

Abbreviations: CI, confidence interval; CNS, central nervous system; MI, myocardial infarction; SE, systemic embolism.

(90.3%) were underdosed (receiving 10 mg once daily despite a CrCl $\geq$50 mL/min) and 278 patients (9.7%) were overdosed (being treated with 15 mg once daily despite a CrCl <50 mL/min). CrCl was not available for 838 patients and one patient received a dose other than 10 mg or 15 mg.

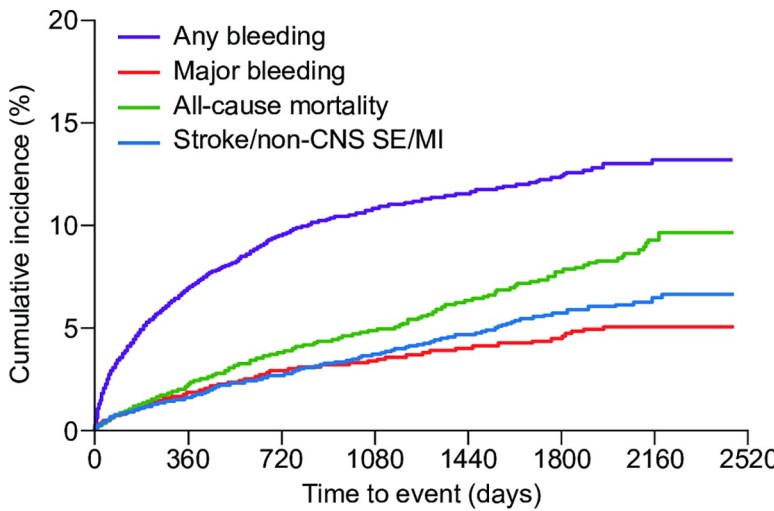

**Fig 2. The cumulative incidences of any bleeding, major bleeding, all-cause mortality, and the composite effectiveness outcome.** The incidences of any bleeding, major bleeding, all-cause mortality, and the composite effectiveness outcome of stroke/non-CNS SE/MI were 3.77, 1.16, 1.77, and 1.32 events per 100 patient-years, respectively. Abbreviations: CNS, central nervous system; MI, myocardial infarction; SE, systemic embolism.

(a)

| Variable | Univariate analysis, HR (95% CI) | Multivariate analysis, HR (95% CI) | |
|---|---|---|---|
| Age ≥75 years | 1.68 (1.34–2.11) | 1.38 (1.06–1.80) | |
| Female sex | 1.05 (0.83–1.32) | 0.94 (0.71–1.22) | |
| Weight ≤50 kg | 1.38 (1.05–1.81) | 1.10 (0.79–1.54) | |
| CrCl <50 mL/min | 1.87 (1.47–2.38) | 1.48 (1.11–1.99) | |
| Congestive heart failure | 1.46 (1.15–1.86) | 1.26 (0.97–1.62) | |
| Hypertension | 1.28 (0.96–1.69) | 1.08 (0.80–1.46) | |
| Diabetes mellitus | 1.45 (1.14–1.85) | 1.45 (1.12–1.86) | |
| Prior ischemic stroke/TIA | 1.46 (1.14–1.88) | 1.27 (0.97–1.65) | |
| Vascular disease[a] | 2.15 (1.42–3.26) | 1.71 (1.11–2.64) | |
| Liver dysfunction | 1.30 (0.88–1.93) | 1.37 (0.91–2.06) | |
| Concomitant use of antiplatelet(s) | 1.89 (1.23–2.89) | 1.55 (0.99–2.44) | |

0.25          1          4
HR (95% CI)

(b)

| Variable | Univariate analysis, HR (95% CI) | Multivariate analysis, HR (95% CI) | |
|---|---|---|---|
| Age ≥75 years | 2.22 (1.78–2.77) | 1.83 (1.42–2.36) | |
| Female sex | 1.00 (0.80–1.24) | 0.85 (0.66–1.09) | |
| Weight ≤50 kg | 1.54 (1.20–1.97) | 1.26 (0.93–1.72) | |
| CrCl <50 mL/min | 1.93 (1.54–2.41) | 1.27 (0.97–1.67) | |
| Congestive heart failure | 1.37 (1.09–1.72) | 1.24 (0.97–1.57) | |
| Hypertension | 1.45 (1.10–1.91) | 1.34 (1.00–1.80) | |
| Diabetes mellitus | 1.14 (0.90–1.45) | 1.09 (0.85–1.41) | |
| Prior ischemic stroke/TIA | 2.16 (1.74–2.69) | 1.72 (1.37–2.17) | |
| Vascular disease[a] | 2.16 (1.41–3.29) | 1.55 (0.99–2.42) | |
| Liver dysfunction | 1.16 (0.77–1.74) | 1.25 (0.82–1.90) | |
| Concomitant use of antiplatelet(s) | 2.43 (1.70–3.50) | 2.02 (1.38–2.97) | |

0.25          1          4
HR (95% CI)

**Fig 3. Univariate and multivariate Cox regression analyses for major bleeding and for the composite effectiveness outcome.** (a) Univariate and multivariate Cox regression analyses in the overall study population for major bleeding. (b) Univariate and multivariate Cox regression analyses in the overall study population for the composite effectiveness outcome of stroke/non-CNS SE/MI. [a] Vascular disease is defined as MI and/or peripheral artery disease and/or aortic plaque. Abbreviations: CI, confidence interval; CNS, central nervous system; CrCl, creatinine clearance; HR, hazard ratio; MI, myocardial infarction; SE, systemic embolism; TIA, transient ischemic attack.

In patients who started on an appropriate dose of rivaroxaban, the incidences of bleeding (major, non-major, and any bleeding) and of the primary composite effectiveness outcome of stroke/non-CNS SE/MI (S6 Table) were comparable with those in the overall study population (Table 2). As in the overall study population, multivariate Cox regression analysis demonstrated that age ≥75 years, prior ischemic stroke/TIA, vascular disease, and concomitant use of antiplatelets were independently associated with increased incidences of major bleeding and stroke/non-CNS SE/MI in patients who started on an appropriate dose of rivaroxaban (S2 Fig).

## Discussion

These results from this prospective, observational study, the XAPASS, provide information on the long-term safety and effectiveness of newly prescribed rivaroxaban in a large and varied population of Japanese patients with NVAF treated in real-world clinical practice over a mean follow-up period of 2.5 years.

In the previous analysis of 1-year outcomes in the XAPASS, the incidences of any and major bleeding were 7.6 and 1.8 events per 100 patient-years, respectively, and the incidence of stroke/non-CNS SE/MI was 1.8 events per 100 patient-years [11]. During up to 5 years of follow-up in the present analysis, the rates of these adverse events remained similarly low at 3.77, 1.16, and 1.32 events per 100 patient-years for any bleeding, major bleeding, and stroke/non-CNS SE/MI, respectively.

There were significant differences between the patient population studied in the XAPASS and the patient population enrolled in the Japanese phase 3 clinical trial Japanese Rivaroxaban Once-Daily Oral Direct Factor Xa Inhibition Compared with Vitamin K Antagonism for Prevention of Stroke and Embolism Trial in Atrial Fibrillation (J-ROCKET AF; NCT00494871) [9, 10]. In the XAPASS, a total of 10,664 patients with NVAF and newly prescribed rivaroxaban were studied, with a third of these patients having CHADS$_2$ scores of 0 (8.4% [900/10,664]) or 1 (24.4% [2601/10,664]). In J-ROCKET AF, the far smaller population of patients with NVAF receiving rivaroxaban (n = 639) had CHADS$_2$ scores ≥2; patients with a lower risk of stroke were excluded from the trial [9]. In addition, in the XAPASS compared with J-ROCKET, the proportions of patients who were female (38.1% vs 17%) and patients aged ≥75 years (48.7% vs 39%) were higher and the frequencies of comorbidities such as congestive heart failure (25.2% vs 41%) and diabetes mellitus (22.8% vs 39%) were lower. Prior ischemic stroke/TIA was reported for 23.0% of patients in the XAPASS, while 64% of patients in J-ROCKET AF had a history of ischemic stroke/TIA/SE. Another important difference between the studies was that a significant proportion of patients in the XAPASS (2594/7295; 35.6%) were underdosed (received 10 mg once daily despite a CrCl ≥50 mL/min) while all patients received the recommended dose in the controlled clinical trial. The incidence of major bleeding was 1.16 events per 100 patient-years in the XAPASS and reported as 3.0% per year in J-ROCKET AF [9]. For the composite outcome of stroke/non-CNS SE, the incidence was 1.19 events per 100 patient-years in the XAPASS and 1.3% per year in J-ROCKET AF.

The real-world safety and effectiveness of rivaroxaban in patients with NVAF in Japan were also evaluated in the investigator-led Evaluation of Effectiveness and Safety of Xa Inhibitor for the Prevention of Stroke and Systemic Embolism in a Nationwide Cohort of Japanese Patients Diagnosed as Non-valvular Atrial Fibrillation (EXPAND) study (NCT02147444). Patient characteristics were similar for the 7141 patients in EXPAND and the XAPASS population; as in the XAPASS, patients with CHADS$_2$ scores of 0 and 1 were included and a significant proportion of patients (30.2%) were underdosed in EXPAND [13, 14]. Results from EXPAND showed that the rivaroxaban dosing regimen recommended in Japan clinical practice for patients with NVAF was associated with low crude incidences of stroke/SE (1.0% per year)

and major and non-major bleeding (1.2% and 4.9% per year, respectively) [13], in agreement with the present analysis. One difference between the studies is that only 1740 patients (24.4%) in EXPAND were newly prescribed rivaroxaban, compared with all 10,664 patients in the XAPASS [13]. The EXPAND and XAPASS studies together provide a significant body of real-world evidence for the safety and effectiveness of rivaroxaban in Japanese patients with NVAF, with the XAPASS providing the majority of data for new users of rivaroxaban.

Outside Japan, the safety and effectiveness of rivaroxaban in the real-world setting is being investigated in the Xarelto for Prevention of Stroke in Patients with Atrial Fibrillation (XANTUS) program. In a pooled analysis of three prospective studies in the XANTUS program that encompassed data from 47 countries from Europe, Canada, Israel, East Asia, the Middle East, Africa, and Latin America and involved a total of 11,121 patients, rivaroxaban was associated with low incidence of major bleeding (1.7 events per 100 patient-years) and stroke/SE (1.0 events per 100 patient-years) in the first year after initiating rivaroxaban treatment [15]. Although the dosing regimen recommended and implemented in the majority of these countries (20 mg once daily in patients with CrCl $\geq$50 mL/min and 15 mg once daily in patients with CrCl <50 mL/min) is different from that in Japan, these studies provide further support for the benefits and safety of rivaroxaban in real-world clinical practice. Furthermore, indirect comparison of the present results with these global data indicate that the incidence of major bleeding in clinical practice in Japan is relatively low (1.16 vs 1.7 events per 100 patient-years in the XAPASS and XANTUS, respectively).

As expected, and previously reported for EXPAND, the incidences of bleeding and the composite effectiveness outcome were higher in patients with higher $CHADS_2$, $CHA_2DS_2$-VASc, or modified HAS-BLED scores at baseline [13]. In multivariate analyses, age $\geq$75 years, renal impairment (CrCl <50 mL/min), diabetes mellitus, and vascular disease were independently associated with an increased incidence of major bleeding in the XAPASS. Age and CrCl were also found to be independently associated with major bleeding in a sub-analysis of EXPAND [16]. In this sub-analysis of EXPAND, only a history of stroke was associated with the stroke/SE outcome in multivariate analysis, while in the present study, age $\geq$75 years, hypertension, and concomitant use of antiplatelets, in addition to prior stroke/TIA, were independently associated with stroke/non-CNS SE/MI. All of these factors warrant careful consideration during assessment of the potential risks of stroke and bleeding in high-risk patients.

A total of 2594 patients (35.6%) started rivaroxaban therapy at the lower dose of 10 mg once daily despite having a CrCl $\geq$50 mL/min and therefore being eligible for the 15 mg once daily dose, in accordance with the product label in Japan. This is most likely because of concerns regarding the risk of bleeding. We had previously reported in an analysis of 1-year XAPASS data that among patients with CrCl $\geq$50 mL/min, those who received an underdose of rivaroxaban were at higher risk than those who received an appropriate dose [17]. That analysis showed that, among patients with CrCl $\geq$50 mL/min, the incidence of major bleeding was similar in underdosed patients compared with those receiving the recommended dose and that the incidence of stroke/non-CNS SE/MI was significantly higher in patients who were underdosed. In this analysis of 2.5 years of follow-up data from the XAPASS, the incidences of bleeding and stroke/non-CNS SE/MI in patients who received the recommended dose of rivaroxaban were almost identical to those in the overall study population. These results provide further support for use of the recommended dosage of rivaroxaban to ensure the optimal balance between safety and effectiveness is achieved in Japanese clinical practice.

## Study limitations

Limitations of the XAPASS include those inherent to the open-label, single-arm, observational design. As an open-label study, selection bias is possible; patients were enrolled if they were

prescribed rivaroxaban, which was determined at the physicians' discretion. The observational design also meant that all decisions regarding management were in fact made at the physicians' discretion; therefore, laboratory and other investigations could not be controlled. The lack of a comparative arm in the study precludes any direct, robust comparisons with other treatments, such as warfarin or other DOACs. Furthermore, it might be difficult to generalize the results of this study to other countries or ethnicities, or to AF patients with high-risk scores for stroke or systemic embolism, since all patients enrolled were Japanese, and the study included mainly patients with low- to intermediate-risk $CHADS_2$ scores. In addition to these factors associated with the study design, the relatively high rate of loss of patients to follow-up may have affected the results and led to an underestimation of incidence. Regarding the high loss rate during follow-up, this may have occurred because the XAPASS was performed as post-marketing surveillance mandated by the Japanese regulatory authority, and was designed as follow-up for up to 5 years. The relatively high proportion of patients who were underdosed in this study may have affected the overall results too. The key strengths of the XAPASS are that it is one of the largest AF registries in Japan and has a prospective design; therefore, it is a source of important, clinically relevant, real-world evidence and complements the available clinical trial data.

## Conclusion

XAPASS has provided information on the safety and effectiveness of rivaroxaban, at reduced doses compared with the dosage used in other countries, in a large number and broad range of patients with NVAF in real-world clinical practice in Japan for 2.5 years of follow-up. No unexpected safety or effectiveness concerns were detected, and the incidences of bleeding and thromboembolic events were low, indicating a favorable balance between safety and effectiveness using the Japanese recommended dosage of rivaroxaban in routine clinical practice.

## Supporting information

**S1 Checklist.**
(PDF)

**S1 Table. Incidences of safety outcomes in patient subgroups.**
(DOCX)

**S2 Table. Adverse events leading to death in the XAPASS.**
(DOCX)

**S3 Table. All adverse events during the standard observation period in the XAPASS.**
(DOCX)

**S4 Table. Incidences of effectiveness outcomes in patient subgroups.**
(DOCX)

**S5 Table. Characteristics of patients in the safety analysis population who started rivaroxaban treatment at the recommended dose.**
(DOCX)

**S6 Table. Safety and effectiveness outcomes in patients who started rivaroxaban treatment at the recommended dose.**
(DOCX)

**S1 Fig.** The incidences of major bleeding and the primary composite effectiveness outcome of stroke, non-CNS SE and MI by baseline $CHADS_2$ score (a), $CHA_2DS_2$-VASc score (b), and

modified HAS-BLED score (c).
(PDF)

**S2 Fig.** Univariate and multivariate Cox regression analyses for major bleeding (a) and primary composite effectiveness outcome of stroke, non-central nervous system systemic embolism, and myocardial infarction (b) in patients who started rivaroxaban treatment at the recommended dose.
(PDF)

**S1 Appendix. Details of the steering committee members for the study.**
(DOCX)

**S1 File.**
(PDF)

**S2 File.**
(PDF)

## Acknowledgments

The authors acknowledge the EPS Corporation for data management and analysis. Medical writing support was provided by Michael Riley, PhD, of PharmaGenesis Cardiff, Cardiff, UK.

## Author Contributions

**Formal analysis:** Toshiyuki Sunaya.

**Investigation:** Yuji Murakawa.

**Methodology:** Takanori Ikeda, Satoshi Ogawa, Takanari Kitazono, Jyoji Nakagawara, Kazuo Minematsu, Susumu Miyamoto, Yuji Murakawa, Yutaka Okayama.

**Project administration:** Yutaka Okayama, Kazufumi Hirano.

**Supervision:** Takanori Ikeda, Satoshi Ogawa, Takanari Kitazono, Jyoji Nakagawara, Kazuo Minematsu, Susumu Miyamoto, Yuji Murakawa.

**Writing – original draft:** Takanari Kitazono, Sanghun Iwashiro.

**Writing – review & editing:** Takanori Ikeda, Satoshi Ogawa, Jyoji Nakagawara, Kazuo Minematsu, Susumu Miyamoto, Yuji Murakawa, Sanghun Iwashiro, Yutaka Okayama, Toshiyuki Sunaya, Kazufumi Hirano, Takanori Hayasaki.

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
