## [Decision Letter · Decision Letter 0]

18 Jan 2021

PONE-D-20-28208

Real-world safety and effectiveness of rivaroxaban using Japan-specific dosage during long-term follow-up in patients with atrial fibrillation: XAPASS

PLOS ONE

Dear Dr. Ikeda,

Thank you for submitting your manuscript to PLOS ONE. After careful consideration, we feel that it has merit but does not fully meet PLOS ONE’s publication criteria as it currently stands. Therefore, we invite you to submit a revised version of the manuscript that addresses the points raised during the review process.

Recommend minor revision, please address the reviewers comments if you would like to resubmit.

Please submit your revised manuscript in 60 days. If you will need more time than this to complete your revisions, please reply to this message or contact the journal office at plosone@plos.org. Please include the following items when submitting your revised manuscript:

We look forward to receiving your revised manuscript.

Kind regards,

Timir Paul

Academic Editor

PLOS ONE

Journal Requirements:

"XAPASS is a post-marketing surveillance study funded by Bayer Yakuhin, Ltd. (Osaka, Japan; https://byl.bayer.co.jp/). A steering committee (S1 Appendix) was responsible for developing the protocol and the case report form and for oversight of both the conduct of the study and the database, and is accountable for analysis and publication of the results. Operational oversight of the study was provided by Bayer Yakuhin, Ltd."

We note that one or more of the authors have an affiliation to the commercial funders of this research study: "Bayer Yakuhin, Ltd.".

2.1. Please provide an amended Funding Statement declaring this commercial affiliation, as well as a statement regarding the Role of Funders in your study. If the funding organization did not play a role in the study design, data collection and analysis, decision to publish, or preparation of the manuscript and only provided financial support in the form of authors' salaries and/or research materials, please review your statements relating to the author contributions, and ensure you have specifically and accurately indicated the role(s) that these authors had in your study. You can update author roles in the Author Contributions section of the online submission form.

2.2. Please also provide an updated Competing Interests Statement declaring this commercial affiliation along with any other relevant declarations relating to employment, consultancy, patents, products in development, or marketed products, etc.  

Reviewers' comments:

Reviewer's Responses to Questions

**Comments to the Author**

1. Is the manuscript technically sound, and do the data support the conclusions?

Reviewer #1: Yes

Reviewer #2: Yes

2. Has the statistical analysis been performed appropriately and rigorously? 

Reviewer #1: Yes

Reviewer #2: I Don't Know

3. Have the authors made all data underlying the findings in their manuscript fully available?

Reviewer #1: Yes

Reviewer #2: Yes

4. Is the manuscript presented in an intelligible fashion and written in standard English?

Reviewer #1: Yes

Reviewer #2: Yes

5. Review Comments to the Author

Reviewer #1: The authors here studies the safety and effectiveness of rivaroxaban using Japan-specific dosage during long-term follow up in patients with non-valvular atrial fibrillation in an open label, single arm observational study (XAPASS). The authors have previously published one year data of safety and effectiveness of rivaroxaban and now following up these patients up to 5 years demonstrating prolonged low adverse events per 100-patient years. The authors also compared it to other observational study (EXPAND) showing similar results. The rationale and careful nature of the data review and statistical methods appear reasonably sound. As pointed out by the authors, the main limitations are loss of follow up of up to 23% patients and lack of comparison arm. The use of rivaroxaban appears safe and effective with reduced doses in Japanese population based on these observational studies.

Reviewer #2: #1: Methods: Please mention inclusion and exclusion criteria

#2 : Study limitations: Please mention about lack of generalized results as limited to only Japanese and also limited to low to intermediate risk CHADVASc score.

#3: Elaborate the reasons why do you have similar efficacy and safety issues with usage of low dose comparted to recommended dose? Also, it seems like most of the patients have low to intermediate CHADVASs score which could be the limitation for real world practice.

#4: Overall, it is a good prospective study with a long term results and authors have done a great job!

6. PLOS authors have the option to publish the peer review history of their article (what does this mean?). If published, this will include your full peer review and any attached files.

Reviewer #1: No

Reviewer #2: No

---

## [Author Response · Author response to Decision Letter 0]

23 Apr 2021

We thank the academic editor and reviewers for reviewing our manuscript. We are grateful for the positive evaluation and very insightful comments. We have revised the manuscript in accordance with the reviewers’ comments.

Subsequent to the submission some errors in the data analyses were found by the external company that performed the analyses following discussion with the Japanese regulatory authority. These errors were mainly among the baseline comorbidity data and do not affect the study conclusion. The manuscript and tables have been amended accordingly, with the changes tracked. The supporting tables and figures for inclusion as Supplemental data have also been amended. 

Academic Editor

1. Comment: Please ensure that your manuscript meets PLOS ONE's style requirements, including those for file naming.

Response: The manuscript has been revised the ensure that it meets PLOS ONE’s style requirements. Please let us know if further revision is needed.

2. Comment: Please provide an amended Funding Statement declaring this commercial affiliation, as well as a statement regarding the Role of Funders in your study. If the funding organization did not play a role in the study design, data collection and analysis, decision to publish, or preparation of the manuscript and only provided financial support in the form of authors' salaries and/or research materials, please review your statements relating to the author contributions, and ensure you have specifically and accurately indicated the role(s) that these authors had in your study. You can update author roles in the Author Contributions section of the online submission form.

Response: The Funding Statement has been amended. This is provided in the cover letter and will be updated in the resubmission.

3. Comment: Please also provide an updated Competing Interests Statement declaring this commercial affiliation along with any other relevant declarations relating to employment, consultancy, patents, products in development, or marketed products, etc.

Response: An updated Disclosure statement is provided in the Cover letter and will be included with our resubmission. 

4. Comment: In your Data Availability statement, you have not specified where the minimal data set underlying the results described in your manuscript can be found. PLOS defines a study's minimal data set as the underlying data used to reach the conclusions drawn in the manuscript and any additional data required to replicate the reported study findings in their entirety. All PLOS journals require that the minimal data set be made fully available. Upon re-submitting your revised manuscript, please upload your study’s minimal underlying data set as either Supporting Information files or to a stable, public repository and include the relevant URLs, DOIs, or accession numbers within your revised cover letter. Any potentially identifying patient information must be fully anonymized.

Important: If there are ethical or legal restrictions to sharing your data publicly, please explain these restrictions in detail. Note that it is not acceptable for the authors to be the sole named individuals responsible for ensuring data access.

Response: We have stated the ethical and legal reasons not to share our data publicly in the Data Availability statement.

 

Reviewer #1

1. Comment: The authors here studies the safety and effectiveness of rivaroxaban using Japan-specific dosage during long-term follow up in patients with non-valvular atrial fibrillation in an open label, single arm observational study (XAPASS). The authors have previously published one year data of safety and effectiveness of rivaroxaban and now following up these patients up to 5 years demonstrating prolonged low adverse events per 100-patient years. The authors also compared it to other observational study (EXPAND) showing similar results. The rationale and careful nature of the data review and statistical methods appear reasonably sound. As pointed out by the authors, the main limitations are loss of follow up of up to 23% patients and lack of comparison arm. The use of rivaroxaban appears safe and effective with reduced doses in Japanese population based on these observational studies.

Response: We appreciate your positive comments. This comment has made us fully recognize the importance of our data for Japanese physicians and patients.

 

Reviewer #2

1. Comment: Methods: Please mention inclusion and exclusion criteria

Response: Thank you for your comment. Eligibility criteria are already stated in Methods – Patients section as follows: Men and Women with NAVAF who were starting rivaroxaban therapy to reduce the risk of stroke/SE were included in the study. Contraindications to rivaroxaban were considered according to the Japanese product label.

2. Comment: Study limitations: Please mention about lack of generalized results as limited to only Japanese and also limited to low to intermediate risk CHADVASc score.

Response: The following statement (underscored) has been added to the Study limitations section of the Discussion: “The lack of a comparative arm in the study precludes any direct, robust comparisons with other treatments, such as warfarin or other DOACs. Furthermore, it might be difficult to generalize the results of this study to other countries or ethnicities, or to AF patients with high-risk scores for stroke or systemic embolism, since all patients enrolled were Japanese, and the study included mainly patients with low- to intermediate-risk CHADS2 scores.” In addition to these factors associated with the study design, the relatively high rate of loss of patients to follow-up may have affected the results and led to an underestimation of incidence.

3. Comment: Elaborate the reasons why do you have similar efficacy and safety issues with usage of low dose comparted to recommended dose? Also, it seems like most of the patients have low to intermediate CHADVASs score which could be the limitation for real world practice.

Response: Thank you for your suggestion, but we have not shown the effectiveness and safety outcomes in patients who received inappropriately low dose of rivaroxaban in this manuscript. Therefore, we would prefer not to include a detailed explanation.

4. Comment: Overall, it is a good prospective study with a long term results and authors have done a great job!

Response: We appreciate your positive comment. It gives us confidence in our study, and we thank the reviewers once again for this thoughtful review.

---

## [Editor Report · Decision Letter 1]

26 Apr 2021

Real-world safety and effectiveness of rivaroxaban using Japan-specific dosage during long-term follow-up in patients with atrial fibrillation: XAPASS

PONE-D-20-28208R1

Dear Dr. Ikeda,

We’re pleased to inform you that your manuscript has been judged scientifically suitable for publication and will be formally accepted for publication once it meets all outstanding technical requirements.

Kind regards,

Timir Paul

Academic Editor

PLOS ONE

Additional Editor Comments (optional):

Reviewers' comments:

All reviewers' comments have been addressed.

---

## [Editor Report · Acceptance letter]

4 Jun 2021

PONE-D-20-28208R1 

Real-world safety and effectiveness of rivaroxaban using Japan-specific dosage during long-term follow-up in patients with atrial fibrillation: XAPASS 

Dear Dr. Ikeda:

I'm pleased to inform you that your manuscript has been deemed suitable for publication in PLOS ONE. Congratulations! Your manuscript is now with our production department. 

Kind regards, 

on behalf of

Dr. Timir Paul 

Academic Editor

PLOS ONE